# PT-MoE: An Efficient Finetuning Framework for Integrating Mixture-of-Experts into Prompt Tuning

**Zongqian Li**
University of Cambridge
zl510@cam.ac.uk

**Yixuan Su**
University of Cambridge
ys484@cam.ac.uk

**Nigel Collier**
University of Cambridge
nhc30@cam.ac.uk

## Abstract

Parameter-efficient fine-tuning (PEFT) methods have shown promise in adapting large language models, yet existing approaches exhibit counter-intuitive phenomena: integrating either matrix decomposition or mixture-of-experts (MoE) individually decreases performance across tasks, though decomposition improves results on specific domains despite reducing parameters, while MoE increases parameter count without corresponding decrease in training efficiency. Motivated by these observations and the modular nature of PT, we propose PT-MoE, a novel framework that integrates matrix decomposition with MoE routing for efficient PT. Evaluation results across 17 datasets demonstrate that PT-MoE achieves state-of-the-art performance in both question answering (QA) and mathematical problem solving tasks, improving F1 score by 1.49 points over PT and 2.13 points over LoRA in QA tasks, while improving mathematical accuracy by 10.75 points over PT and 0.44 points over LoRA, all while using 25% fewer parameters than LoRA. Our analysis reveals that while PT methods generally excel in QA tasks and LoRA-based methods in math datasets, the integration of matrix decomposition and MoE in PT-MoE yields complementary benefits: decomposition enables efficient parameter sharing across experts while MoE provides dynamic adaptation, collectively enabling PT-MoE to demonstrate cross-task consistency and generalization abilities. These findings, along with ablation studies on routing mechanisms and architectural components, provide insights for future PEFT methods. [1]

## 1 Introduction

**Background.** Large language models (LLMs) have shown remarkable capabilities but require resources for fine-tuning [41, 18, 19]. PEFT methods address this challenge by updating only a small subset of parameters [8, 21, 22]. **Prompt tuning** (PT) stands out among PEFT approaches with its unique advantages: minimizing trainable parameters through soft prompt optimization, enabling modular deployment through task-specific prompts without model modifications, and supporting flexible knowledge composition [14, 20]. These properties make it particularly effective for low-resource and multi-task applications where efficient adaptation is essential [17].

**Motivation.** Despite these advantages, we observe three **counter-intuitive phenomena** in prompt tuning. **First**, applying either matrix decomposition or MoE routing individually leads to performance decrease in both QA and mathematical tasks compared to standard PT (PT vs DPT, PT vs SMoP; Figure 1). **Second**, matrix decomposition, while reducing parameter count, improves performance on specific subsets of tasks (PT vs DPT; Tables 2, 3, 4), revealing task-dependent optimization dynamics. **Third**, integrating additional routing components or multiple experts increases parameter count without corresponding decrease in training efficiency (PT vs SMoP, LoRA vs HydraLoRA; Figure 4). These phenomena indicate that the relationship between parameter efficiency and model

---

[1] https://github.com/ZongqianLi/PT-MoE

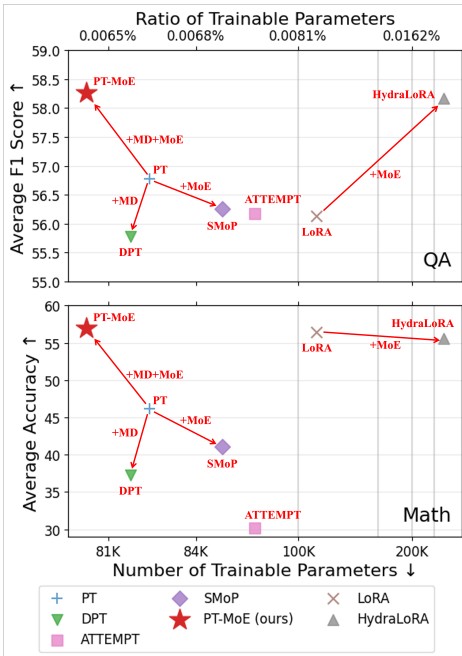

Figure 1: **Performance and parameter efficiency comparison** of PEFT methods on QA and mathematical tasks. The upper subgraph shows average F1 scores on 12 MRQA benchmark datasets, while the lower subgraph shows average accuracy on 5 mathematical datasets. The x-axis represents the number of trainable parameters, with corresponding parameter ratio shown at the top. ↑ indicates higher is better; ↓ indicates lower is better. Red arrows indicate method transformations: +MD (matrix decomposition), +MoE (mixture-of-experts), or their combination. PT excels in QA tasks while LoRA demonstrates advantages in mathematical tasks. PT-MoE achieves the best performance on both task types while using fewer parameters than alternative methods, demonstrating that combining matrix decomposition and MoE yields complementary benefits despite each component individually decreasing performance when applied to PT.

effectiveness in prompt tuning is more nuanced than previously understood, motivating the need for a more sophisticated approach to prompt optimization.

Based on these observations, we propose a novel framework, **Prompt Tuning with Efficient Mixture-of-Experts (PT-MoE)**, that combines matrix decomposition with MoE routing. As shown in Figure 1, our approach not only achieves state-of-the-art performance, but also keeps modular and uses minimal trainable parameters and moderate training steps.

**Contributions.** Our work offers three key developments:

- **Novel finetuning framework:** We propose PT-MoE, integrating matrix decomposition with MoE for prompt tuning. Our framework achieves state-of-the-art performance with fewer parameters while outperforming either technique alone, demonstrating their complementary benefits.
- **Design dynamics:** We analyze key variables influencing the performance of PT-MoE, including prompt length, expert count, routing mechanisms, and model size. Our findings provide design guidelines for future parameter-efficient tuning approaches.
- **Key insights:** Our comprehensive analysis across diverse tasks reveals several important findings: **First**, prompt tuning methods excel in QA tasks while LoRA-based methods demonstrate advantages in mathematical reasoning; **Second**, matrix decomposition reduces parameters while potentially improving domain-specific performance, whereas MoE integration increases parameter count without compromising training efficiency; and **Third**, combining matrix decomposition and MoE enables PT-MoE to achieve superior performance across all tasks while maintaining minimal parameter count and moderate training costs, whereas applying either of them individually can decrease average performance.

**Organization.** The remainder of this paper is organized as follows: Section 2 reviews related work in prompt tuning, covering both direct tuning approaches and transfer learning methods. Section 3 presents our PT-MoE framework, detailing the matrix decomposition strategy, dynamic router design, and training methodology. Section 4 describes our experimental design across QA and mathematical problem-solving tasks. Section 5 presents comprehensive results, including detailed ablation studies analyzing the influences of prompt length, parameter count, expert number, routing mechanisms, and model size, followed by efficiency analysis. Section 6 concludes with key findings and future directions.

## 2 Related Work

To contextualize our approach, we review existing prompt tuning methods, which fall into two categories: direct prompt tuning approaches focusing on architectural innovations, and transfer learning methods enabling cross-task knowledge sharing.

**Direct prompt tuning** methods have evolved into four main branches: (1) General approaches that directly optimize prompt parameters, including Prompt Tuning that prepends trainable vectors to input while freezing the language model [14], XPrompt that employs pruning to identify and retain important prompt tokens [25], and P-Tuning v2 that introduces deep prompts across all transformer layers [23]; (2) Encoder-based methods that leverage additional additional modules, such as P-Tuning that incorporates an encoder to learn dependencies between continuous embeddings [24], Residual Prompt Tuning (RPT) that employs a residual part with down/up-projection layers for stable optimization [31], and Prefix Tuning that prepends trainable key-value pairs at each layer through a reparameterization section [16]; (3) Decomposition methods that decompose prompt embeddings, including Decomposed Prompt Tuning (DPT) that applies low-rank matrix decomposition to reduce parameter count [37], and DePT that combines shorter soft prompts with low-rank updates to word embeddings [33]; and (4) MoE approaches such as Sparse Mixture-of-Prompts (SMoP) that employs multiple shorter prompts with a dynamic gating mechanism to route inputs to the most suitable prompt representations [2].

**Transfer learning** approaches in prompt tuning have developed into three categories: (1) General approaches that directly transfer prompt knowledge, including SPoT that introduces both generic transfer through multi-task pre-training and targeted transfer via task similarity matching [34], and ATTEMPT that dynamically combines multiple source prompts through an attention-based mixing mechanism with instance-level adaptation [1]; (2) Encoder-based methods that facilitate knowledge transfer through additional architectures, such as TransPrompt that employs parallel task-specific and universal encoders with balancing mechanisms for obtaining both task-dependent and task-agnostic knowledge [35], and Cross-Task Prompt Tuning (CTPT) that leverages multi-head attention for cross-task knowledge transfer with dimension reduction and derivative-free optimization [38]; and (3) Decomposition methods exemplified by Multitask Prompt Tuning (MPT) that decomposes prompts into shared and task-specific components through knowledge distillation, enabling efficient transfer while preserving task-specific adaptability through a rank-one decomposition strategy [36].

## 3 Methods

Building upon the insights from prior work, we propose a new parameter-efficient prompt tuning framework, PT-MoE, shown in Figure 2 and Algorithm 1.

**Framework Overview.** PT-MoE integrates matrix decomposition and dynamic routing. Given an input sequence $\mathbf{x}$, our framework first generates routing weights $\mathbf{w}$ through a router network $R$: $\mathbf{w} = R(\mathbf{x})$. These weights determine the selection among $N$ decomposed prompts, where each prompt $\mathbf{P}_i$ is decomposed as $\mathbf{P}_i = \mathbf{A}_i\mathbf{B}$, with $\mathbf{A}_i$ being prompt-specific and $\mathbf{B}$ being shared across all prompts. The final soft prompt $\mathbf{P}$ is computed as $\mathbf{P} = \sum_{i=1}^{N} w_i\mathbf{A}_i\mathbf{B}$, which is then prepended to the input sequence for the frozen language model.

**Matrix Decomposition.** To achieve parameter efficiency, we decompose each prompt matrix $\mathbf{P}_i \in \mathbb{R}^{T \times H}$ into a prompt-specific matrix $\mathbf{A}_i \in \mathbb{R}^{T \times R}$ and a shared matrix $\mathbf{B} \in \mathbb{R}^{R \times H}$, where $T$, $H$, and $R$ denote the prompt length, hidden dimension, and low-rank dimension respectively. This reduces parameters from $O(NTH)$ to

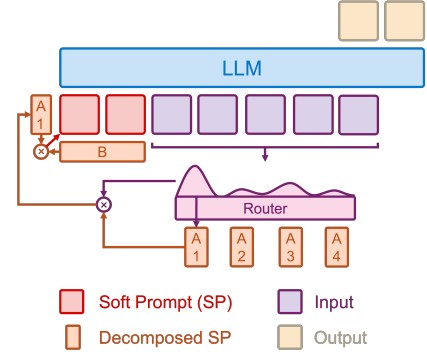

Figure 2: **Architecture** of PT-MoE. Each soft prompt is decomposed into an input-specific matrix $A_i$ and a shared matrix $B$, with a router adaptively selecting and combining prompt components based on input. The resulting soft prompt is prepended to the input for the frozen LLM.

---

**Algorithm 1** Pseudo code of PT-MoE

---

**Require:** Base model $\mathcal{M}$; input batch $X = x_1, \ldots, x_b$; parameters $\theta$
  **Notation:** $b$ - batch size; $s$ - sequence length; $n$ - number of prompts; $k$ - tokens per prompt; $d$ - low-rank dimension; $h$ - hidden dimension
  **for** batch $x \in X$ **do**
      Get input embeddings $E = \mathcal{M}_{\text{embed}}(x)$ *where* $E \in \mathbb{R}^{b \times s \times h}$
      Calculate mean embeddings $\mu = \text{mean}(E, \dim = 1)$ *where* $\mu \in \mathbb{R}^{b \times h}$
      Compute router logits $l = W\mu + b$ *where* $W \in \mathbb{R}^{n \times h}, b \in \mathbb{R}^n, l \in \mathbb{R}^{b \times n}$
      Get router weights $w = \text{softmax}(l)$ *where* $w \in \mathbb{R}^{b \times n}$
      **for** each sample $j$ in batch **do**
         Find indices of top-k weights: $i_{topk} = \text{argsort}(w_j)[-k :]$
         Zero all weights except top-k: $w_j[i] = 0$ for all $i \notin i_{topk}$
      **end for**
      Initialize prompt embeddings $P = 0, P \in \mathbb{R}^{b \times k \times d}$
      **for** each weight $w_i$ in $w$ **do**
         Compute weighted prompts $P = P + w_i A_i$ *where* $A_i \in \mathbb{R}^{k \times d}$
      **end for**
      Project to model dimension $P = P \times B$ *where* $B \in \mathbb{R}^{d \times h}$
      Combine with input: $C = \text{concat}(P, E)$ *where* $C \in \mathbb{R}^{b \times (k+s) \times h}$
      Generate through base model: $y = \mathcal{M}(C)$
  **end for**
**Ensure:** Model predictions $y$

---

| **MRQA (Extractive QA)** | |
|---|---|
| In-domain | SQuAD [30], TriviaQA [9], SearchQA [5], HotpotQA [39], NaturalQuestions [12] |
| Out-of-domain | BioASQ [28], DROP [4], DuoRC [32], RACE [13], RelationExtraction [15], TextbookQA [10] |
| **Mathematics (Problem Solving)** | |
| In-domain | GSM8K [3] |
| Out-of-domain | SVAMP: Subtraction, Addition, Common-Division, Multiplication [29]; ASDIV [26]; MAWPS [11]; MATH_PROBLEMS [27] |

Table 1: Overview of training and evaluation **datasets**. The experiments span two task categories: extractive QA (MRQA benchmark with 12 QA datasets) and mathematical problem solving (GSM8K and specialized mathematical datasets). For each category, datasets are divided into in-domain sets used for training, validation, and evaluation, and out-of-domain sets used exclusively for testing generalization capability.

$O(NTR + RH)$ for $N$ prompts. The low-rank dimension $R$ is either manually specified or automatically computed to maintain parameter efficiency. For initialization, we first encode task-relevant text to obtain embeddings $\mathbf{E} \in \mathbb{R}^{T \times H}$, then perform SVD: $\mathbf{E} = \mathbf{U\Sigma V}^\top$. Each $\mathbf{A}_i$ is initialized as $\mathbf{U}$: $R\Sigma R^{1/2}$ and the shared $\mathbf{B}$ as $\mathbf{\Sigma} R^{1/2} \mathbf{V}_{R:}^\top$, where subscript $R$ indicates truncation to the first $R$ components. This approach ensures the initial prompts encode task-relevant information while maintaining the parameter efficiency of the decomposition.

**Dynamic Router.** The router network adaptively selects prompts based on input context. Given an input sequence embedding $\mathbf{x} \in \mathbb{R}^H$ (obtained by averaging token embeddings), the router computes logits through a linear projection: $\mathbf{l} = \mathbf{Wx} + \mathbf{b}$, where $\mathbf{W} \in \mathbb{R}^{N \times H}$ and $\mathbf{b} \in \mathbb{R}^N$. During training, we apply multiplicative Gaussian noise to encourage exploration: $\mathbf{l}' = \mathbf{l} \odot (1 + \epsilon)$, where $\epsilon \sim \mathcal{N}(0, \sigma^2)$. The routing weights are computed as $\mathbf{w} = \text{softmax}(\mathbf{l}') \odot \mathbf{1}_{\text{argmax}}$, where $\mathbf{1}_{\text{argmax}}$ is a one-hot vector with 1 at the position of the maximum value. This hard selection strategy reduces interference between prompts while maintaining end-to-end differentiability through straight-through estimation.

**Training and Prediction.** During training, we optimize both the router parameters and decomposed prompt matrices while keeping the base model frozen. For language model training, we use negative log-likelihood loss computed only on non-prompt positions using a binary mask: $\mathcal{L} = -\sum_{t \in \mathcal{M}} \log p(y_t | x_{<t})$, where $\mathcal{M}$ denotes non-prompt positions. We employ AdamW optimizer with warmup followed by a constant learning rate schedule, and gradient accumulation for stable optimization. At inference, noise is not added in the router, ensuring deterministic prompt selection.

## 4 Experimental Design

**Datasets.** We conduct evaluations across **17** diverse datasets, as shown in Table 1, where in-domain datasets are split into training, validation, and test sets, while out-of-domain datasets are

used exclusively for testing. For **QA**, we utilize 12 MRQA datasets [6], with in-domain sets like SQuAD [30] testing information extraction abilities and out-of-domain sets like DROP [4] evaluating domain adaptation. For mathematical **problem solving**, we use GSM8K [3] from MetaMath [40] as our in-domain benchmark, complemented by specialized out-of-domain datasets including the subject-specific subsets of SVAMP [29], ASDIV [26], MAWPS [11], and MATHPROBLEMS [27].

**Gold Standard and Baselines.** We employ full model fine-tuning as our **gold standard**, which updates all parameters but requires computational resources. Our **baselines**[2] include representative methods from **prompt tuning categories**: For **direct prompt tuning**, we select (1) PT from general approaches, (2) DPT from decomposition methods, and (3) SMoP from MoE approaches. While **transfer learning** methods like (4) ATTEMPT typically involve multi-turn training, we also evaluate its architecture under similar training for comprehensive comparison. We additionally compare against **other PEFT methods** including (5) LoRA and (6) HydraLoRA, with HydraLoRA adopting a MoE-like architecture that uses a shared down-projection matrix and multiple routed up-projection matrices. **These two LoRA-based methods require model architecture modifications unlike the modular nature of prompt tuning methods.**

**Evaluation Metrices.** We employ task-specific evaluation metrics. For extractive QA tasks from MRQA, we adopt two metrics: **F1** score, which evaluates the token-level overlap between predicted and ground truth answer spans, balancing precision and recall; and **Exact Match (EM)**, which measures the percentage of predictions that exactly match the ground truth. For mathematical problem solving tasks, we use **accuracy**, defined as the percentage of correctly solved problems with exact answer matches.

**Models.** We conduct our main experiments using **LLaMA-3.2-1B-Instruct** as the base model for fine-tuning methods [7]. For ablation studies on model size, we additionally employ **LLaMA-3.2-3B-Instruct**.

## 5 Results

### 5.1 Question Answering

The experimental results on MRQA datasets shown in Table 2 and 3 demonstrate the effectiveness of PT-MoE across various QA tasks. We highlight seven key findings: (1) PT-MoE achieves superior overall performance with an average F1 score of 58.26%, outperforming SMoP (56.25%) by 2.01 points and the standard PT (56.77%) by 1.49 points, establishing a new state-of-the-art on the MRQA benchmark. (2) This advancement is further validated by Exact Match metrics, where PT-MoE demonstrates even more gains (47.13% for average, surpassing SMoP and PT by 2.16 and 1.61 points respectively). (3) PT-MoE exhibits strong generalization capabilities across both in-domain and out-of-domain scenarios. It achieves the highest performance on four out of six in-domain datasets and three out of six out-of-domain datasets. (4) The stability of PT-MoE is evidenced by consistent improvements over PT across 11 out of 12 datasets, with only marginal decreases in the RACE dataset. In contrast, SMoP shows performance decrease on 5 datasets compared to PT. (5) **Individual architectural components show limited gains:** both matrix decomposition (DPT, 55.77% F1) and MoE (SMoP, 56.25% F1) underperform standard prompt tuning (PT, 56.77% F1). (6) **PT-MoE's integration of matrix decomposition and MoE yields complementary benefits, outperforming both DPT and SMoP by 2.49 and 2.01 points for F1 respectively. This improvement over individual approaches proves the mutually beneficial nature of these techniques.** (7) Notably, while PT-MoE achieves lower overall performance than FT, it reaches comparable or even higher scores than FT on specific datasets such as DROP (48.02% vs 43.87% F1) while using only 80K parameters compared to FT's 1.2B. These results collectively validate the effectiveness of the architectural design of PT-MoE and demonstrate its superior performance in accuracy and generalization across diverse QA scenarios.

### 5.2 Mathematical Problem Solving

The experimental results on mathematical tasks (Table 4) reveal several distinctive characteristics compared to QA tasks. We highlight six key findings: (1) PT-MoE achieves state-of-the-art per-

---

[2]All methods are controlled to have similar parameter budgets, with detailed configurations shown in Table 9 of the Appendix.

| | FT | LoRA | HydraLoRA | PT | DPT | SMoP | ATTEMPT | PT-MoE |
|---|---|---|---|---|---|---|---|---|
| # para. | 1.2B | 106k | 278k | 81k | 81k | 86k | 90k | **80k** |
| **In-domain** | | | | | | | | |
| SQ | 78.76 | 69.82 | 74.24 | 72.31 | 70.99 | 74.15 | **74.22** | 73.85 |
| News | 48.69 | 39.91 | 44.05 | 48.18 | 48.42 | **48.96** | 48.18 | 48.24 |
| Tri | 71.04 | 70.61 | 71.38 | 65.93 | 65.41 | 66.13 | 65.31 | **67.34** |
| Srch | 71.35 | 55.56 | 60.13 | 49.74 | 46.94 | 41.08 | 37.64 | **51.33** |
| HP | 72.96 | 63.29 | 64.02 | 58.69 | 58.49 | 58.96 | 60.18 | **62.16** |
| NQ | 67.56 | 65.92 | 66.31 | 62.18 | 61.65 | 61.17 | 59.59 | **62.95** |
| **Out-of-domain** | | | | | | | | |
| BSQ | 70.19 | 65.38 | 68.76 | 68.59 | 65.56 | 68.59 | 66.69 | **69.33** |
| DP | 43.87 | 35.25 | 34.38 | 40.39 | 38.80 | 39.92 | 45.32 | **48.02** |
| DRC | 48.11 | 43.69 | 44.36 | 43.30 | 43.64 | 42.07 | 42.86 | **43.96** |
| RC | 43.44 | 38.04 | 40.00 | 42.10 | 41.89 | 42.34 | **43.01** | 42.51 |
| RE | 81.60 | 74.09 | 77.97 | 82.43 | 80.85 | 83.73 | **84.11** | 83.70 |
| TB | 52.71 | 52.00 | 52.44 | 47.34 | 46.62 | **47.85** | 46.91 | 45.71 |
| Avg. | 62.52 | 56.13 | 58.17 | 56.77 | 55.77 | 56.25 | 56.17 | **58.26** |

Table 2: Evaluation results (F1 scores) for various PEFT methods on **QA** datasets. SQ: SQuAD; News: NewsQA; Tri: TriviaQA; Srch: SearchQA; HP: HotpotQA; NQ: NaturalQuestions; BSQ: BioASQ; DP: DROP; DRC: DuoRC; RC: RACE; RE: RelationExtraction; TB: TextbookQA. The bold values indicate the best performance among prompt tuning-based methods.

| | FT | LoRA | HydraLoRA | PT | DPT | SMoP | ATTEMPT | PT-MoE |
|---|---|---|---|---|---|---|---|---|
| # para. | 1.2B | 106k | 278k | 81k | 81k | 86k | 90k | **80k** |
| **In-domain** | | | | | | | | |
| SQ | 65.28 | 56.26 | 61.63 | 61.25 | 58.49 | 63.15 | **63.71** | 63.34 |
| News | 32.76 | 25.26 | 27.80 | 32.62 | **32.88** | 32.81 | 32.50 | 32.85 |
| Tri | 62.29 | 64.11 | 64.32 | 59.49 | 58.56 | 59.48 | 58.71 | **60.87** |
| Srch | 61.50 | 46.10 | 50.06 | 42.40 | 39.65 | 34.51 | 31.24 | **43.98** |
| HP | 56.19 | 47.48 | 47.73 | 44.45 | 44.33 | 43.80 | 45.77 | **47.29** |
| NQ | 50.45 | 49.54 | 49.59 | 47.28 | 46.54 | 46.39 | 45.66 | **48.18** |
| **Out-of-domain** | | | | | | | | |
| BSQ | 49.06 | 42.02 | 44.01 | 51.79 | 49.46 | 50.06 | 49.26 | **52.06** |
| DP | 32.26 | 25.48 | 24.75 | 30.60 | 28.74 | 29.94 | 36.06 | **37.12** |
| DRC | 38.84 | 33.24 | 33.57 | 34.64 | 35.64 | 34.11 | 34.84 | **35.64** |
| RC | 29.52 | 24.92 | 26.11 | 29.82 | 30.26 | 30.56 | 30.41 | **31.75** |
| RE | 66.99 | 58.58 | 62.68 | 72.45 | 70.48 | 74.59 | **75.13** | 74.18 |
| TB | 43.71 | 44.17 | 43.97 | 39.52 | 38.72 | **40.25** | 39.52 | 38.25 |
| Avg. | 49.07 | 43.09 | 44.69 | 45.52 | 44.48 | 44.97 | 45.23 | **47.13** |

Table 3: Evaluation results (Exact Match) for **QA** datasets.

formance with an average accuracy of 56.91%, improving upon PT (46.16%) by 10.75 points, demonstrating its effectiveness in mathematical reasoning. (2) **The benefits of MoE integration show method-dependent characteristics**: in prompt tuning approaches, PT-MoE and SMoP demonstrate different changes over PT (by +10.75 and -5.11 points respectively); when applied to LoRA methods, HydraLoRA shows slightly performance decrease compared to LoRA. (3) **LoRA-based methods demonstrate advantages in mathematical tasks compared to their performance in QA. While LoRA underperformed PT by 5.36 points in MRQA, it outperforms PT by 10.31 points in mathematical tasks, indicating task-specific strengths of different PEFT approaches.** (4) **PT-MoE demonstrates unique cross-task consistency:** while prompt tuning methods excel in QA tasks and LoRA-based methods in mathematical tasks, PT-MoE achieves the highest average performance in both domains, indicating robust adaptability across different problem types. (5) While PEFT methods consistently underperform full fine-tuning, the performance gap is larger in mathematical tasks compared to QA tasks, with a wider performance range among different methods. Notably, PT-MoE achieves comparable or higher performance to full fine-tuning on specific datasets such as Division and MP500. (6) PT-MoE demonstrates superior parameter efficiency, achieving higher performance than LoRA while using only 75% of its parameters (80k vs 106k), and outperforming HydraLoRA which uses 3.5 times more parameters. These findings highlight both the unique challenges of mathematical tasks and the robust adaptability of PT-MoE across different problem domains.

## 5.3 Case Study

To better understand the performance characteristics of PT-MoE, we present a detailed case study of polynomial addition in Table 5. In this example, the response of the base model suffers from

| | FT | LoRA | HydraLoRA | PT | DPT | SMoP | ATTEMPT | PT-MoE |
|---|---|---|---|---|---|---|---|---|
| # para. | 1.2B | 106k | 278k | 81k | 81k | 86k | 90k | **80k** |
| **In-domain** | | | | | | | | |
| GSM8K | 58.15 | 41.77 | 41.31 | 34.11 | 26.08 | 27.97 | 27.36 | **35.63** |
| **Out-of-domain** | | | | | | | | |
| Sub. | 68.75 | 67.50 | 57.50 | 41.87 | 43.12 | 38.12 | 40.00 | **55.62** |
| Add. | 64.40 | 61.01 | 62.71 | 50.84 | 35.59 | 35.59 | 35.59 | **55.93** |
| Div. | 62.50 | 52.08 | 52.08 | 66.66 | 64.58 | 33.33 | 37.50 | **79.16** |
| Multi. | 48.48 | 33.33 | 39.39 | 33.33 | 27.27 | 33.33 | 27.27 | **36.36** |
| SVAMP | 61.03 | 53.48 | 52.92 | 48.18 | 42.64 | 35.09 | 35.09 | **56.77** |
| ASDIV | 86.04 | 73.42 | 74.08 | 60.13 | 56.14 | 49.50 | 24.91 | **77.74** |
| MAWPS | 82.53 | 70.70 | 76.05 | 57.18 | 43.09 | 65.91 | 49.01 | **71.83** |
| MP500 | 30.60 | 43.00 | 33.40 | 31.20 | 18.20 | 26.80 | 14.60 | **42.60** |
| Avg. | 63.67 | 56.47 | 55.55 | 46.16 | 37.23 | 41.05 | 30.19 | **56.91** |

Table 4: Accuracy (%) on **mathematical problem-solving** tasks with the number of trainable parameters shown in the second row. The first four out-of-domain datasets are from the SVAMP dataset: Sub. (Subtraction), Add. (Addition), Div. (Common-Division), and Multi. (Multiplication). MP500 denotes the first 500 questions from MATH_PROBLEMS.

**Question:** Let $f(y) = y^4 - 3y^3 + y - 3$ and $g(y) = y^3 + 7y^2 - 2$. Find $f(y) + g(y)$. Write your answer as a polynomial with terms of decreasing degree. **Standard Answer:** $y^4 - 2y^3 + 7y^2 + y - 5$

**Original:**
## Step 1: Identify the polynomials f(y) and g(y)
The polynomial f(y) is given as $f(y) = y^4 - 3y^3 + y - 3$ and the polynomial g(y) is given as $g(y) = y^3 + 7y^2 - 2$.
## Step 2: Add the polynomials f(y) and g(y)
To find the sum of f(y) and g(y), we need to add the corresponding terms of each polynomial.
## Step 3: Perform the addition
$f(y) + g(y) = (y^4 - 3y^3 + y - 3) + (y^3 + 7y^2 - 2)$
## Step 4: Combine like terms
Combine the terms with the same degree to simplify the expression.
## Step 5: Simplify the expression
$f(y) + g(y) = y^4 - 3y^3 + y^3 + 7y^2 - 2 - 3$
## Step 6: Combine like terms
Combine the terms with the same degree.
## Step 7: Simplify the expression
$f(y) + g(y) = y^4 - 2y^3 + 7y^2 - 2$
## Step 8: Write the final answer
The final answer is: $y^4 - 2y^3 + 7y^2 - 2$

**Prompt Tuning:**
## Step 1: To find the sum of the two polynomials $f(y)$ and $g(y)$, we need to add their corresponding terms.
## Step 2: The polynomial $f(y) = y^4 - 3y^3 + y - 3$ has terms of degree 4, 3, 2, and 1, while the polynomial $g(y) = y^3 + 7y^2 - 2$ has terms of degree 3, 2, and 1.
## Step 3: When we add these two polynomials, we need to combine like terms, which means adding the coefficients of terms with the same degree.
## Step 4: The terms with degree 4 are $y^4$ and $y^4$, so their sum is $2y^4$.
## Step 5: The terms with degree 3 are $-3y^3$ and $-3y^3$, so their sum is $-6y^3$.
## Step 6: The terms with degree 2 are $y$ and $7y^2$, so their sum is $7y^2$.
## Step 7: The terms with degree 1 are 1 and $-2$, so their sum is $-1$.
## Step 8: Therefore, the sum of the two polynomials $f(y)$ and $g(y)$ is $2y^4 - 6y^3 + 7y^2 - 1$.
## Step 9: The answer is $2y^4 - 6y^3 + 7y^2 - 1$.

**PT-MoE:**
## Step 1: Identify the polynomials f(y) and g(y)
The given polynomials are $f(y) = y^4 - 3y^3 + y - 3$ and $g(y) = y^3 + 7y^2 - 2$.
## Step 2: Add the polynomials f(y) and g(y)
To find $f(y) + g(y)$, we need to add the corresponding terms of the two polynomials.
## Step 3: Perform the addition
$f(y) + g(y) = (y^4 - 3y^3 + y - 3) + (y^3 + 7y^2 - 2)$
## Step 4: Combine like terms
Combine the terms with the same degree of y:
$y^4 + (y^3 - 3y^3) + (y^3 + 7y^2) + (y - 3) - 2$
## Step 5: Simplify the expression
Simplify the expression by combining the coefficients of like terms: $y^4 - 2y^3 + 7y^2 + y - 5$
## Step 6: Write the final expression
The expression $f(y) + g(y)$ is $y^4 - 2y^3 + 7y^2 + y - 5$.
## Note: $y$ is the variable, and the answer is $y^4 - 2y^3 + 7y^2 + y - 5$.

Table 5: A **case study** comparing three solution approaches (Original, Prompt Tuning, and PT-MoE) for a polynomial addition problem. Errors in the outputs are highlighted in red (incorrect terms), orange (missing terms), and blue (hallucinated terms).

information loss, specifically omitting the linear term during simplification steps, leading to an incorrect final result. The conventional prompt tuning approach exhibits hallucinations and conceptual errors, particularly in degree identification and term combination, resulting in spurious terms like $2y^4$ and $-6y^3$. PT-MoE maintains information integrity throughout the solution process and avoids hallucinations, ultimately producing the correct polynomial expression. Notably, PT-MoE achieves this with a more concise solution structure, demonstrating efficient problem-solving steps while maintaining accuracy.

## 5.4 Ablation Studies

To systematically evaluate the design choices in PT-MoE, we conduct ablation studies on **five influencing variables**: soft prompt length, trainable parameters, number of experts, routing mechanisms, and model size. For each experiment, we keep other variables fixed at their default values (soft prompt length=40, trainable parameters≈80K, number of experts=2, probationary-selective routing, 1B base model) while varying the target component to isolate its influences on model performance.

**Soft prompt length.** We evaluate prompt lengths ranging from 20 to 80 tokens (Figure 3 Left). Three consistent observations emerge: (1) In-domain performance exceeds out-of-domain across all lengths, maintaining a 5-6% F1 score margin; (2) Both domains achieve optimal performance at 40 tokens, with peak F1 scores of 60.66% and 55.28% respectively; and (3) Performance in both domains follows a similar trajectory, improving up to 40 tokens then declining. These findings

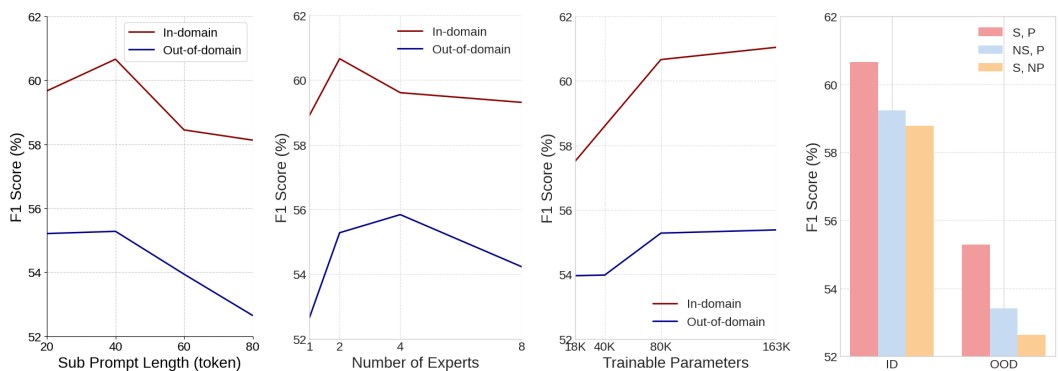

Figure 3: **Ablation studies** on key components of PT-MoE, showing the influences of (Left) prompt length, (Center left) number of experts, (Center right) trainable parameters, and (Right) routing mechanisms ((N)S: (Non-)Selective, (N)P: (Non-)Probationary) on in-domain (ID) and out-of-domain (OOD) performance.

indicate that the optimal prompt length is domain-agnostic, though the absolute performance levels remain domain-dependent.

**Number of experts.** We investigate the influences of expert count by varying it from 1 to 8 (Figure 3 Center left). There are three key points: (1) Single-expert configuration yields the poorest performance (58.90% and 52.64% F1 for in-domain and out-of-domain), demonstrating the necessity of MoE; (2) Performance exhibits an initial increase followed by decrease, with in-domain peaking at N=2 (60.66% F1) and out-of-domain at N=4 (55.84% F1), suggesting different optimal routing capacities for each domain; (3) In-domain tasks consistently outperform out-of-domain scenarios by a 4-6% F1 margin across all expert counts. These observations demonstrate that the optimal number of experts varies by domain type and highlight the importance of balancing expert specialization with routing complexity.

**Trainable parameters.** We vary the parameter count from 18K to 163K to analyze its influence on model performance (Figure 3 Center Right). Three key observations emerge: (1) Performance consistently improves with increasing parameters, from 57.51% to 61.04% F1 for in-domain and 53.96% to 55.38% F1 for out-of-domain tasks, and notably maintains stability even at higher parameter counts, contrasting with conventional prompt tuning methods; (2) While both in-domain and out-of-domain tasks show positive scaling, they exhibit distinct parameter sensitivity behaviours, in-domain tasks demonstrate rapid improvement before 80K parameters, while out-of-domain tasks show accelerated growth in the 40K-80K range; (3) In-domain performance maintains a consistent advantage over out-of-domain tasks across all parameter settings, with F1 scores differing by approximately 4-6%. These findings suggest that PT-MoE effectively leverages additional parameters to achieve continuous performance gains.

**Routing mechanisms.** We examine two key routing design choices (Figure 3 Right): selective routing, which activates only the highest-weighted expert versus non-selective routing that utilizes all experts with their respective weights, and probationary routing, which scales the output by the router's selection probability versus non-probationary routing that uses unscaled outputs. Our experiments reveal four key findings: (1) The combination of selective and probationary routing (S, P) consistently outperforms other configurations (NS, P and S, NP) across both in-domain (60.66% vs 59.24% and 58.78% F1) and out-of-domain tasks (55.28% vs 53.41% and 52.64% F1), suggesting the complementary benefits of focused expert utilization and confidence-based output scaling; (2) Probationary routing demonstrates superior performance over its non-probationary counterpart, indicating the value of incorporating router confidence in the final output; (3) Under probationary conditions, selective routing achieves 1.42% higher F1 score while reducing active parameters compared to non-selective routing, highlighting the effectiveness and efficiency of specialized expert knowledge; (4) All routing configurations maintain higher performance on in-domain tasks compared to out-of-domain scenarios, though the relative performance remains consistent across domains. These findings collectively demonstrate that the selective probationary routing mechanism achieves an optimal balance between model performance and computational efficiency.

**Model size.** We conduct additional experiments using a 3B version of the base model, comparing PT-MoE with PT and the MoE-integrated method, SMoP (Table 6). Three key findings emerge: (1) PT-MoE maintains its competitiveness at larger scales, achieving the highest average accuracy of 71.11%, surpassing standard PT (66.43%) and SMoP (69.61%). (2) SMoP shows scale-dependent behavior: while underperforming PT on the 1B model (56.77% vs 56.25%), it surpasses PT on the 3B model (69.61% vs 66.43%). (3) PT-MoE demonstrates robust performance by outperforming the baselines on three

| | PT | SMoP | PT-MoE |
|---|---|---|---|
| GSM8K | 56.70 | **61.78** | 59.74 |
| SVAMP | 69.36 | **74.69** | 72.81 |
| ASDIV | 76.41 | 80.06 | **81.39** |
| MAWPS | 70.70 | 70.70 | **78.02** |
| MP500 | 59.00 | 60.80 | **63.60** |
| Average | 66.43 | 69.61 | **71.11** |

Table 6: Performance comparison (accuracy %) of standard and MoE-based prompt tuning methods on mathematical problem solving tasks using a **3B** base model.

out of five mathematical datasets. These findings collectively validate the scalability and stability of PT-MoE across different model sizes.

## 5.5 Efficiency Analysis

Results in Figure 4 demonstrate PT-MoE's efficiency across both computational and parametric dimensions. PT-MoE achieves the highest performance with only moderate training steps and minimal parameters (80k). In contrast, LoRA and HydraLoRA require more parameters and training steps to achieve comparable performance. Other prompt tuning methods such as PT, SMoP, and DPT converge fast but achieve lower performance, suggesting a potential trade-off between training efficiency and model effectiveness. This evidence validates that PT-MoE balances the computational cost, parameter efficiency, and model performance.

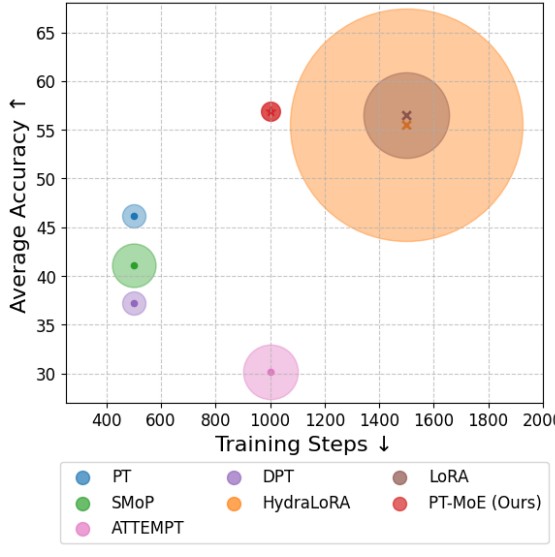

Figure 4: Parameter and training **efficiency comparison** across different methods. The x-axis shows training steps for the highest performance after training parameter search, while the y-axis shows the average accuracy on math datasets. Circle sizes indicate the number of trainable parameters, with larger circles representing more parameters. Circle markers represent modular methods (PT-based) that preserve model architecture, while x markers indicate LoRA-based methods requiring model structure modifications. Though PT-MoE introduces a router, it's implemented as a shallow layer with negligible inference overhead compared to generation time.

## 6 Conclusions

This work introduces PT-MoE, a novel parameter-efficient framework that integrates matrix decomposition with MoE routing for prompt tuning. Our experiments across 16 datasets demonstrate that PT-MoE achieves state-of-the-art performance while maintaining parameter efficiency, outperforming existing methods in both QA and mathematical tasks. Through ablation studies, we identify optimal configurations for prompt length, expert count, and routing mechanisms, providing insights for future parameter-efficient tuning approaches.

Future directions include using different routers to manage different task, and extending PT-MoE to continual learning scenarios for efficient adaptation and knowledge transfer across tasks.

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

# A  Appendix

## A.1  Implementation Details

We provide implementation details, including training hyperparameters in Table 8, inference parameters in Table 7, and method-specific configurations in Tables 9 to facilitate reproducibility.

|  | QA | Math |
| --- | --- | --- |
| max_new_tokens | 100 | 768 |
| num_beams | 1 | 1 |
| do_sample | False | False |
| temperature | 1.0 | 1.0 |
| top_p | 1.0 | 1.0 |
| pad_token_id | pad_token_id | pad_token_id |
| eos_token_id | eos_token_id | eos_token_id |
| early_stopping | True | True |

Table 7: **Inference parameters** for QA and mathematical tasks.

|  | QA | Math |
| --- | --- | --- |
| Train steps | {500, 1000, 1500} for PT-based methods {200, 600, 1000} for LoRA-based methods | {500, 1000, 1500} |
| Optimizer | AdamW | AdamW |
| Max length | 512 | 768 |
| warmup_steps | 500 | 500 |
| learning_rate | 2e-5 | 2e-5 |
| per_device_train_batch_size | 32 | 16 |
| lr_scheduler_type | constant_with_warmup | constant_with_warmup |
| gradient_accumulation_steps | 2 | 2 |

Table 8: **Training hyperparameters** for QA and mathematical tasks. {} means parameter search.

| Method | Details |
| --- | --- |
| LoRA | r=1; lora_alpha=16; target_modules=["q_proj", "v_proj"]; lora_dropout=0; bias="none"; task_type=TaskType.CAUSAL_LM |
| HydraLoRA | r=1; alpha=16; target_modules=["q_proj", "v_proj"]; dropout=0.0; num_b_matrices=2; Router: nn.Sequential(nn.Linear(input_dim, num_b_matrices)); Initialization: A: nn.init.kaiming_uniform_(, a=math.sqrt(5)), B: nn.init.zeros_() |
| PT | Soft prompt length: 40; Initialization: Specific words |
| DPT | Soft prompt length: 40; low_rank_dim = 39; Initialization: Specific words; Decomposition method: SVD |
| SMoP | Total soft prompt length: 40; Number of experts: 2; Initialization: Specific words; Noise: *(1+torch.randn_like()*0.01) |
| ATTEMPT | Total soft prompt length: 40; Number of experts: 2; Encoder: nn.Linear(embedding_dim, projection_dim=1), nn.Linear(projection_dim=1, embedding_dim), nn.LayerNorm(embedding_dim); Initialization: Specific words |
| PT-MoE | Soft prompt length: 40; Number of expert: 2; Rank: 36; Router: nn.Linear(embedding_dim, num_prompts); Noise: *(1+torch.randn_like()*0.01) |

Table 9: **Method configurations** for various PEFT methods.

Prompt structure for MRQA:

```
<|start_header_id|>user<|end_header_id|>\n\nExtract the exact text span from
the given context that directly answers the question, without modifying or
combining multiple parts of the text.\n\nContext: {}\n\nQuestion: {}<|eot_id
|><|start_header_id|>assistant<|end_header_id|>\n\nAnswer:
```

Prompt structure for Math datasets:

```
<|start_header_id|>user<|end_header_id|>\n\nSolve the question and your
response should end with \"The answer is: [answer]\".\n\nQuestion: {}<|eot_id
|><|start_header_id|>assistant<|end_header_id|>\n\nAnswer:
```

Texts used to initialize soft prompt for finetuning on MRQA:

```
(
    "Read the following context and answer the question. "
    "Extract the answer from the context. "
    "The answer is a span of the context."
    "Answer the question directly."
    "Use the original words in the context."
    "Do not introduce any words not present in the context."
)
```

Texts used to initialize soft prompt for finetuning on Math datasets:

```
(
    "Read the question carefully and make sure you understand it before
    beginning. "
    "Pay close attention to the details and requirements of the question. "
    "Answer the question, ensuring your response is relevant to what is asked
    . "
    "Ensure your answer is both accurate and correct."
)
```

## A.2 Environment

```
python==3.11.5
torch==2.3.1+cu118
transformers==4.46.0
datasets==2.18.0
huggingface_hub==0.24.2
deepspeed==0.15.3
wandb==0.14.2
numpy==1.23.5
tqdm==4.66.4
```

