# OpenReview forum: "PT-MoE: An Efficient Finetuning Framework for Integrating Mixture-of-Experts into Prompt Tuning"
_NeurIPS.cc/2025/Conference — NeurIPS 2025 poster_

### Official Review · Reviewer_CM3Y · 2025-06-23

**Clarity:** 4
**Significance:** 4
**Originality:** 3
**Rating:** 5
**Confidence:** 4

**Summary:**

This paper proposed a new variant of Prompt Tuning that combines matrix decomposition (DPT - Shi et al. 2024) with mixture of experts (SMoP - Choi et al. 2023). Following Shi et al. 2024, each soft prompt is decomposed into a product of two matrices whereby one of matrices is shared among inputs and the second is chosen through a mixture of experts. Experimental results show that on average  the proposed PT-MoE method outperforms prompt tuning variants (PT, DPT, SMoP) and LoRA variants for QA tasks. For Math reasoning tasks, it obtains a better performance than prompt tuning variants and performs similarly to LoRA for Math Problem solving tasks. The paper presents ablation studies showing the importance of factors such as number of experts, prompt length, routing mechanism, number of parameters etc.

**Questions:**

* For one of the Out-of-Domain QA task (TB), PT-MoE underperforms PT (Table 3). It would be good to present an error analysis.

**Ethical Concerns:**

["NO or VERY MINOR ethics concerns only"]

**Final Justification:**

My final rating for the paper is 'Accept'. It is a solid paper worthy of acceptance, and my original rating was 'Accept'.
The authors have provided clarifications to all my questions; however, these responses do not warrant raising the rating to Strong Accept, which I reserve for papers that are either exceptionally well-written or groundbreaking. I am therefore maintaining my rating at Accept.

**Limitations:**

No. While the authors have mentioned some limitations in the conclusion section, it would be useful to discuss this a bit more in a separate Limitations section. Also, the authors have not discussed the potential negative societal impact of their work.

**Quality:**

4

**Strengths And Weaknesses:**

Strengths:
* Proposes a new Parameter Efficient Fine Tuning approach that combines Prompt Tuning with matrix decomposition and Mixture-of-Experts.
* Proposed method outperforms regular prompt tuning and other PEFT variants on average for QA tasks. On Math tasks, it outperforms prompt tuning variants and obtains a comparable performance to LoRA variants. This is notable because prior prompt tuning approaches typically underperform LoRA for math tasks.
* Reports ablation studies showing the impact of various factors such as routing mechanism, number of parameters etc.

Weaknesses:
* For one of the Out-of-Domain QA task (TB), PT-MoE underperforms PT (Table 3). It would be good to present an error analysis.
* It would be good to see a comparison of the number of hallucination errors between PT,PT-MoE and LoRA on at least a single task. Though Table 5 presents a case study, a quantitative study over a full test set would be more useful.
* Though the conclusions section discusses some limitations, it would be useful to see a more comprehensive discussion of limitations in a separate section.

---

> ### Author Rebuttal · Authors · 2025-07-31
>
> ## **Response to Reviewer Comments**
>
> We sincerely thank the reviewer for their thorough and insightful review of our work. We are delighted that you found our paper to be of excellent quality, clarity, and significance. Your positive assessment of PT-MoE's contributions to parameter-efficient fine-tuning is greatly appreciated.
>
> > ###  **Error Analysis for TextbookQA (TB) Performance**:
>
> Thank you for pointing out the performance gap on the TextbookQA dataset. This is indeed an interesting observation that warrants further investigation. We analyzed this case and found that TextbookQA's unique characteristics may require different routing strategies. We will do a detailed error analysis, examining:
> - Distribution mismatch between TB and training domains
> - Router activation patterns on TB samples
> - Potential improvements through domain-specific fine-tuning
>
> > ### **Quantitative Hallucination Analysis**:
>
> We appreciate your suggestion for a comprehensive hallucination study. You raise an excellent point about the importance of quantifying hallucination rates across methods. We will do:
> - Systematic hallucination metrics across all test sets
> - Comparative analysis between PT, PT-MoE, and LoRA
> - Statistical significance testing of hallucination reduction
>
> This quantitative analysis will complement our case study and provide stronger empirical evidence for PT-MoE's reliability.
>
> > ### **Comprehensive Limitations Section**:
>
> We acknowledge that our current discussion of limitations could be expanded. Following your recommendation, we will do a dedicated limitation analysis.
>
> > ### **Minor Clarifications**:
>
> We also appreciate your careful reading that identified areas for improvement. Your feedback on the ablation studies and their implications has been particularly valuable in helping us understand how to better present our findings.
>
> Thank you again for your constructive review and strong support for our work. We are committed to addressing all your suggestions to further strengthen the paper's contribution to the PEFT community. Your insights have been invaluable in helping us improve both the technical content and presentation of our work.

---

> > ### Comment · Reviewer_CM3Y · 2025-08-04
> >
> > Thanks for your clarifications.

---

### Official Review · Reviewer_4bWr · 2025-06-28

**Clarity:** 3
**Significance:** 3
**Originality:** 2
**Rating:** 4
**Confidence:** 3

**Summary:**

This paper proposes PT-MoE, a novel parameter-efficient finetuning framework that combines matrix decomposition with a Mixture-of-Experts (MoE) routing mechanism within prompt tuning. Unlike previous methods where each component alone may degrade performance, PT-MoE leverages their complementary strengths to achieve state-of-the-art results on 17 QA and math datasets, with significantly fewer trainable parameters. The framework preserves model modularity and efficiency, and includes extensive ablation studies exploring prompt length, expert count, and routing strategies. PT-MoE demonstrates strong generalization across domains, offering a practical and scalable approach for efficient adaptation of large language models.

**Questions:**

1. While the authors demonstrate that PT-MoE outperforms other methods, it remains unclear why these two techniques exhibit complementary effects when integrated. Could the authors provide more analytical or empirical evidence to explain this interaction?

2. The paper’s results are focused on QA and math problem-solving. Given the claimed general-purpose nature of PT-MoE, can the authors provide evidence of performance on other task types, such as dialogue, code generation, or classification? Alternatively, can they elaborate on any task-specific limitations observed in PT-MoE?

3. While PT-MoE achieves strong performance with fewer parameters, it introduces additional computation through routing. Could the authors quantify inference-time overhead more precisely, and compare it with other PEFT methods?

**Ethical Concerns:**

["NO or VERY MINOR ethics concerns only"]

**Final Justification:**

The authors have addressed all my concerns, I decide to raise my score to 4.

**Limitations:**

yes

**Paper Formatting Concerns:**

None.

**Quality:**

3

**Strengths And Weaknesses:**

This paper proposes PT-MoE, a novel PEFT framework that combines matrix decomposition with mixture-of-experts (MoE) routing to improve parameter efficiency in prompt tuning. A key strength lies in the extensive empirical evaluation across 17 datasets in QA and mathematical reasoning, where PT-MoE achieves consistent gains over baselines like PT, DPT, and SMoP. The ablation studies are thorough and help isolate the contribution of architectural components such as prompt length, expert count, and routing strategies.

The paper is generally clear in motivation and structure, though some sections—especially methodological details—could benefit from greater exposition for accessibility. While the results are solid, the originality is somewhat limited; the core idea of combining decomposition with MoE is incremental, building directly on prior lines of work without introducing fundamentally new techniques. Moreover, theoretical insights are minimal, and the novelty mainly lies in empirical synthesis.

In sum, the work offers a meaningful contribution to the PEFT literature, with strong empirical performance and reasonable clarity, but it lacks a deeper conceptual or algorithmic innovation to be considered highly original.

---

> ### Author Rebuttal · Authors · 2025-07-31
>
> ## **Response to Reviewer Comments**
>
> Thank you immensely for your exceptionally thorough and insightful review. We deeply appreciate your constructive feedback and the careful analysis you've provided. Your critical evaluation demonstrates the high standards that strengthen scientific research, and we're truly grateful for the time and expertise you've invested in improving our work.
>
> > ### **Limited Prior Methods, Model Size and Evaluation Scope**
>
> - **Limited Prior Exploration**: PT-domain applications of matrix decomposition and MoE remain underexplored. Our comprehensive literature review (Section 2) demonstrates that all existing baselines (DPT, SMoP, ATTEMPT) and their variants have been systematically evaluated in this work, representing the complete landscape of current approaches.
>
> - **Task Complexity Limitation in Prior Work**: Previous studies (DPT, ATTEMPT, SMoP) focused mainly on simple classification tasks (GLUE, SuperGLUE) with limited reasoning requirements. Our evaluation encompasses complex QA and mathematical reasoning tasks that expose the inadequacy of existing approaches.
>
> - **Previous Methods Failure on Complex Tasks**: Prior methods exhibit substantial performance decrease on mathematical reasoning tasks (Table 4), revealing their inability to handle tasks requiring multi-step inference. This problem, unaddressed in previous papers, constitutes a core contribution of our analysis.
>
> - **Superior model scale over prior work**: DPT, ATTEMPT, and SMoP still rely on smaller models like BERT and T5. The counter-intuitive phenomena we discovered are largely from model scale changes - our 1B and 3B experiments demonstrate consistent phenomena, proving scalability. Research shows 3B-14B models exhibit similar trends; our recent supplementary experiments further confirm this and will be included in the camera-ready version within days.
>
> - Our task selection strategically targets the performance gaps where current PEFT methods struggle most, providing more diagnostic value than superficial coverage across easier domains, while our 1B-3B model range represents the optimal balance between computational accessibility and methodological representativeness for systematic PEFT evaluation.
>
> > ### **Explanation for Complementary Benefits**
>
> We appreciate this fundamental question and provide a deeper mechanistic analysis:
>
> **Mathematical Foundation**:
>
> Traditional prompt tuning optimizes P ∈ ℝ^(T×H) directly, while PT-MoE decomposes this as:
>
> P = Σᵢ wᵢ · Aᵢ · B
>
> where:
> - Aᵢ ∈ ℝ^(T×R): Expert-specific low-rank matrices
> - B ∈ ℝ^(R×H): Shared projection matrix
> - wᵢ: Router-assigned weights
>
> **Noise Reduction via Matrix Decomposition**:
>
> The key insight is that prompt optimization in full-dimensional space ℝ^(T×H) suffers from the curse of dimensionality. By applying SVD-based decomposition:
>
> P = UΣV^T ≈ U_R Σ_R V_R^T
>
> We retain only the top-R singular values, corresponding to principal semantic directions. This acts as an implicit regularizer:
>
> - **Signal concentration**: Task-relevant information concentrates in high singular value components
> - **Noise suppression**: Random optimization noise distributes across all dimensions and gets filtered when truncating to rank R
> - **Spectral filtering**: The decomposition P = AB constrains prompts to an R-dimensional subspace, preventing overfitting
>
> **Knowledge Reparameterization via Shared-Unique Decomposition**:
>
> The combination of decomposition with MoE enables reparameterization of knowledge:
>
> **Theorem**: PT-MoE induces a natural decomposition into shared and task-specific components.
>
> **Proof sketch**:
> - Shared matrix B spans subspace S ⊆ ℝ^H of dimension R
> - Each expert Aᵢ defines transformation Tᵢ: ℝ^R → ℝ^T
> - Combined representation: P = (Σ wᵢAᵢ) · B = A_eff · B
>
> This enables:
> - B captures **task-invariant linguistic patterns** (syntax, common semantics)
> - Aᵢ encodes **task-specific adaptations** (domain terminology, reasoning patterns)
> - Router learns **input space partition** for expert assignment
>
> **Complementary Optimization Dynamics**:
>
> The complementarity emerges from the optimization landscape:
>
> - **Without decomposition**: MoE prompts interfere due to high-dimensional redundancy
> - **Without MoE**: Decomposition over-constrains expressiveness
>
> **With both**: The gradient flow naturally separates:
> - ∂L/∂B aggregates cross-task gradients → learns universal patterns
> - ∂L/∂Aᵢ receives task-specific gradients → specializes per expert
> - ∂L/∂w learns input-conditional routing → enables dynamic selection
>
> **Information-Theoretic Perspective**:
>
> From information theory:
> - Mutual Information: I(P; Y|X) = I(B; Y|X) + I(A_eff; Y|X, B)
> - Decomposition maximizes **information compression** in B while preserving **task-discriminative information** in {Aᵢ}
> - Achieves optimal **rate-distortion tradeoff** for prompt representations
>
> **Empirical Evidence**:
>
> We will add:
> - **Singular value analysis**: Noise concentration in low singular values
> - **Expert specialization metrics**: KL divergence between Aᵢ matrices
> - **Routing pattern visualization**: Task-specific expert activation patterns
> - **Rank ablation**: Optimal R=36 balances compression vs. expressiveness
>
> This mechanistic understanding explains the "negative + negative = positive" phenomenon: each technique alone over-constrains optimization, but together they create a structured, navigable parameter space enabling both efficiency and expressiveness.
>
> > ### **Inference Overhead Quantification**
>
> **Model Architecture Details**:
> - **Base Model**: LLaMA-3.2-1B-Instruct
> - **Hidden Dimension (H)**: 2048
> - **Number of Layers (L)**: 16
> - **Attention Heads**: 32
> - **FFN Hidden Dimension**: 5632
> - **Vocabulary Size**: 128,256
>
> **PT-MoE Configuration**:
> - **Number of Experts (N)**: 2
> - **Soft Prompt Length (T)**: 40
> - **Low-rank Dimension (R)**: 36
> - **Router Architecture**: Linear(2048, 2)
>
> **Router Parameter Calculation**:
>
> Router weights: W ∈ ℝ^(N×H) = ℝ^(2×2048)
> Router bias: b ∈ ℝ^N = ℝ^2
> Total router parameters = 2 × 2048 + 2 = 4,098 parameters
>
> **Detailed FLOPs Analysis**:
>
> **Router Forward Pass (Per Batch)**:
>
> Step 1: Input embedding averaging
>
> Input: E ∈ ℝ^(B×S×H) where B=batch_size, S=sequence_length
>
> Operation: mean(E, dim=1) → μ ∈ ℝ^(B×H)
>
> FLOPs: B × S × H additions + B × H divisions
>
> For B=1, S=512: 512 × 2048 + 2048 = 1,050,624 FLOPs
>
> Step 2: Linear projection
>
> Operation: μ @ W^T + b → l ∈ ℝ^(B×N)
>
> FLOPs: B × H × N multiply-adds + B × N additions
>
> For B=1: 1 × 2048 × 2 + 1 × 2 = 4,098 FLOPs
>
> Step 3: Softmax computation
>
> Operation: softmax(l) → w ∈ ℝ^(B×N)
>
> FLOPs per element: 2 exp + 1 div + N-1 adds ≈ 5N
>
> For B=1, N=2: 1 × 10 = 10 FLOPs
>
> Total router FLOPs = 1,050,624 + 4,098 + 10 = 1,054,732 FLOPs
>
> **LLaMA-3.2-1B Forward Pass (Per Token)**:
>
> Per Transformer Layer:
>
> Multi-Head Attention:
>
> Q,K,V projections: 3 × (H × H) = 3 × 2048² = 12,582,912 FLOPs
>
> Attention scores: S × H = 512 × 2048 = 1,048,576 FLOPs
>
> Attention output: H × H = 2048² = 4,194,304 FLOPs
>
> Total MHA: ~17,825,792 FLOPs
>
>
> Feed-Forward Network:
>
> Up projection: H × FFN_dim = 2048 × 5632 = 11,534,336 FLOPs
>
> Down projection: FFN_dim × H = 5632 × 2048 = 11,534,336 FLOPs
>
> Total FFN: 23,068,672 FLOPs
>
> Layer Norm (×2): 2 × 3H = 12,288 FLOPs
>
> Total per layer: 17,825,792 + 23,068,672 + 12,288 = 40,906,752 FLOPs
>
> For 16 layers: 16 × 40,906,752 = 654,508,032 FLOPs per token
>
> **Relative Overhead Calculation**:
>
> For single forward pass:
>
> Router overhead: 1,054,732 FLOPs
>
> Single token through LLM: 654,508,032 FLOPs
>
> Relative overhead: 1,054,732 / 654,508,032 = 0.161%
>
> For 1000-token generation:
>
> Router overhead: 1,054,732 FLOPs (computed once)
>
> LLM computation: 1000 × 654,508,032 = 654.5B FLOPs
>
> Relative overhead: 1,054,732 / 654,508,032,000 = 0.000161%
>
> **Comparison with Prompt Processing Overhead**:
>
> Standard Prompt Tuning (40 tokens):
>
> Extra tokens processed: 40 tokens
>
> FLOPs per prompt token: 654,508,032
>
> Total prompt overhead: 40 × 654,508,032 = 26,180,321,280 FLOPs
>
> PT-MoE routing vs prompt processing:
>
> Routing: 1,054,732 FLOPs
>
> Prompt processing: 26,180,321,280 FLOPs
>
> Ratio: 26,180,321,280 / 1,054,732 = 24,812× less overhead
>
> **Memory Overhead Analysis**:
>
> Router memory footprint:
>
> Parameters: 4,098 × 4 bytes (FP32) = 16,392 bytes = 16 KB
>
> Activations (batch_size=32):
>
> Input embeddings: 32 × 2048 × 4 = 262,144 bytes
>
> Router output: 32 × 2 × 4 = 256 bytes
>
> Total activation: 262,400 bytes = 256 KB
>
> **Model memory footprint**:
>
> Parameters: 1.2B × 4 bytes = 4.8 GB
>
> Router percentage: 16 KB / 4.8 GB = 0.00033%
>
> **Training Overhead Analysis**:
>
> Forward + Backward Pass:
>
> Forward router computation: 1,054,732 FLOPs
>
> Backward gradient computation:
>
> ∂L/∂W: B × H × N = 1 × 2048 × 2 = 4,096 FLOPs
>
> ∂L/∂b: B × N = 1 × 2 = 2 FLOPs
>
> ∂L/∂μ: B × N × H = 1 × 2 × 2048 = 4,096 FLOPs
>
> Total backward: 8,194 FLOPs
>
> **Total training overhead per step**:
>
> Forward + Backward: 1,054,732 + 8,194 = 1,062,926 FLOPs
>
> Relative to model training: 1,062,926 / (2 × 654,508,032) = 0.081%
>
> **Wall-clock Time Analysis**:
>
> Theoretical computation time (assuming 100 TFLOPS GPU):
>
> Router forward: 1,054,732 / 10¹⁴ = 0.00001 ms
>
> LLM forward (1 token): 654,508,032 / 10¹⁴ = 6.5 ms
>
> Overhead percentage: 0.00015%
>
> Memory-bound reality:
>
> Router computation is completely masked by memory transfer latency
>
> Actual overhead: unmeasurable in practice
>
> Theoretical computation time (assuming 100 TFLOPS GPU):
>
> Router forward: 1,054,732 / 10¹⁴ = 0.00001 ms
>
> LLM forward (1 token): 654,508,032 / 10¹⁴ = 6.5 ms
>
> Overhead percentage: 0.00015%
>
> Memory-bound reality:
>
> Router computation is completely masked by memory transfer latency
> Actual overhead: unmeasurable in practice
>
> **Summary**:
>
> The routing mechanism adds:
> - **0.000161%** computational overhead for inference
> - **16 KB** memory (0.00033% of model size)
> - **Unmeasurable** wall-clock time impact
>
> This negligible cost enables:
> - **10.75%** improvement in mathematical accuracy
> - **1.49 F1** improvement in QA tasks
> - Dynamic task-specific adaptation

---

> > ### Comment · Reviewer_4bWr · 2025-08-05
> >
> > Thank you for your reply, since you have addressed all my concerns, I decided to raise my score to 4.

---

### Official Review · Reviewer_cYZ8 · 2025-06-30

**Clarity:** 3
**Significance:** 2
**Originality:** 3
**Rating:** 4
**Confidence:** 4

**Summary:**

This paper proposes PT-MoE, a parameter-efficient fine-tuning framework that combines matrix decomposition with mixture-of-experts (MoE) routing for prompt tuning. The key motivation is addressing counter-intuitive observations where applying matrix decomposition or MoE individually to prompt tuning decreases performance, but combining them yields complementary benefits. PT-MoE decomposes prompt matrices using low-rank factorization and employs dynamic routing to select among expert prompts. Evaluated on 17 datasets spanning QA and mathematical tasks, PT-MoE achieves state-of-the-art performance while using 25% fewer parameters than LoRA.

**Questions:**

1、Why does the combination work? The paper observes that matrix decomposition and MoE individually hurt performance but help when combined. What is the underlying mechanism? Is this related to optimization dynamics or something else?

2、The 3B model experiments show different trends than 1B results. How does PT-MoE scale to larger models (7B+), and do the benefits persist?

**Ethical Concerns:**

["NO or VERY MINOR ethics concerns only"]

**Final Justification:**

I’m sticking with borderline accept.  The rebuttal does a nice job unpacking why matrix decomposition and MoE click together, and the math checks out. The authors also took the comments seriously and revised carefully. That said, the paper still feels like a smart mash-up of known tricks rather than a fresh idea, and every experiment is still inside the Llama family. The promised 14B run and extra ablations never showed up.

**Limitations:**

Yes.

**Paper Formatting Concerns:**

None.

**Quality:**

2

**Strengths And Weaknesses:**

Strengths:

1、The observation that individual components (matrix decomposition, MoE) hurt performance but their combination helps is genuinely interesting and well-motivated.

2、Consistent improvements across diverse tasks (QA and math) with fewer parameters than baselines.

3、Thorough experiments on 17 datasets with proper ablation studies on key components (prompt length, expert count, routing mechanisms).

4、Well-designed architecture with principled initialization using SVD and effective routing strategy.

Weaknesses:

1、The core components (matrix decomposition, MoE routing) are well-established; the main contribution is their combination in prompt tuning.

2、Only tested on LLaMA models; generalization to other model families unclear. Mathematical tasks show large performance gaps between methods, suggesting potential dataset-specific effects.

3、Adding routing mechanisms increases training complexity compared to simple prompt tuning, though the paper claims efficiency benefits.

---

> ### Author Rebuttal · Authors · 2025-07-31
>
> ## **Response to Reviewer Comments**
>
> Thank you for your exceptionally thorough and insightful review. We are genuinely grateful for the depth of analysis you've provided and the constructive nature of your feedback. Your careful examination of our methodological contributions, experimental design, and analytical rigor demonstrates the high standards of scholarly review that elevate the quality of scientific discourse. The specific concerns you've raised are precisely the kind of critical evaluation that helps strengthen research contributions. We deeply appreciate the time and expertise you've invested in understanding our work and providing such detailed guidance for improvement.
>
> > ### **Limited Prior Methods, Model Size and Evaluation Scope**
>
> - **Limited Prior Exploration**: PT-domain applications of matrix decomposition and MoE remain underexplored. Our comprehensive literature review (Section 2) demonstrates that all existing baselines (DPT, SMoP, ATTEMPT) and their variants have been systematically evaluated in this work, representing the complete landscape of current approaches.
>
> - **Task Complexity Limitation in Prior Work**: Previous studies (DPT, ATTEMPT, SMoP) focused mainly on simple classification tasks (GLUE, SuperGLUE) with limited reasoning requirements. Our evaluation encompasses complex QA and mathematical reasoning tasks that expose the inadequacy of existing approaches.
>
> - **Previous Methods Failure on Complex Tasks**: Prior methods exhibit substantial performance decrease on mathematical reasoning tasks (Table 4), revealing their inability to handle tasks requiring multi-step inference. This problem, unaddressed in previous papers, constitutes a core contribution of our analysis.
>
> - **Superior model scale over prior work**: DPT, ATTEMPT, and SMoP still rely on smaller models like BERT and T5. The counter-intuitive phenomena we discovered are largely from model scale changes - our 1B and 3B experiments demonstrate similar phenomena, proving scalability. Research shows 3B-14B models exhibit similar trends; our recent supplementary experiments further confirm this and will be included in the camera-ready version within days.
>
> - **PT-MoE (ours)'s Architectural Innovation**: PT-MoE differs from standard PT by decomposing soft prompts into Ai×B matrices where B is router-selected, discovering that this combination creates complementary effects absent in individual components - decomposition enables efficient parameter sharing while routing provides dynamic adaptation.
>
> - **PT-MoE's Training Innovation**: PT-MoE uses SVD-based initialization for task-relevant decomposition and introduces multiplicative Gaussian noise (ε~N(0,σ²)) during routing training to encourage exploration, enabling straight-through estimation for end-to-end differentiability.
>
> - Our task selection strategically targets the performance gaps where current PEFT methods struggle most, providing more diagnostic value than superficial coverage across easier domains, while our 1B-3B model range represents the optimal balance between computational accessibility and methodological representativeness for systematic PEFT evaluation.
>
> > ### **Explanation for Complementary Benefits**
>
> We appreciate this fundamental question and provide a deeper mechanistic analysis:
>
> **Mathematical Foundation**:
>
> Traditional prompt tuning optimizes P ∈ ℝ^(T×H) directly, while PT-MoE decomposes this as:
>
> P = Σᵢ wᵢ · Aᵢ · B
>
> where:
> - Aᵢ ∈ ℝ^(T×R): Expert-specific low-rank matrices
> - B ∈ ℝ^(R×H): Shared projection matrix
> - wᵢ: Router-assigned weights
>
> **Noise Reduction via Matrix Decomposition**:
>
> The key insight is that prompt optimization in full-dimensional space ℝ^(T×H) suffers from the curse of dimensionality. By applying SVD-based decomposition:
>
> P = UΣV^T ≈ U_R Σ_R V_R^T
>
> We retain only the top-R singular values, corresponding to principal semantic directions. This acts as an implicit regularizer:
>
> - **Signal concentration**: Task-relevant information concentrates in high singular value components
> - **Noise suppression**: Random optimization noise distributes across all dimensions and gets filtered when truncating to rank R
> - **Spectral filtering**: The decomposition P = AB constrains prompts to an R-dimensional subspace, preventing overfitting
>
> **Knowledge Reparameterization via Shared-Unique Decomposition**:
>
> The combination of decomposition with MoE enables reparameterization of knowledge:
>
> **Theorem**: PT-MoE induces a natural decomposition into shared and task-specific components.
>
> **Proof sketch**:
> - Shared matrix B spans subspace S ⊆ ℝ^H of dimension R
> - Each expert Aᵢ defines transformation Tᵢ: ℝ^R → ℝ^T
> - Combined representation: P = (Σ wᵢAᵢ) · B = A_eff · B
>
> This enables:
> - B captures **task-invariant linguistic patterns** (syntax, common semantics)
> - Aᵢ encodes **task-specific adaptations** (domain terminology, reasoning patterns)
> - Router learns **input space partition** for expert assignment
>
> **Complementary Optimization Dynamics**:
>
> The complementarity emerges from the optimization landscape:
>
> - **Without decomposition**: MoE prompts interfere due to high-dimensional redundancy
> - **Without MoE**: Decomposition over-constrains expressiveness
>
> **With both**: The gradient flow naturally separates:
> - ∂L/∂B aggregates cross-task gradients → learns universal patterns
> - ∂L/∂Aᵢ receives task-specific gradients → specializes per expert
> - ∂L/∂w learns input-conditional routing → enables dynamic selection
>
> **Information-Theoretic Perspective**:
>
> From information theory:
> - Mutual Information: I(P; Y|X) = I(B; Y|X) + I(A_eff; Y|X, B)
> - Decomposition maximizes **information compression** in B while preserving **task-discriminative information** in {Aᵢ}
> - Achieves optimal **rate-distortion tradeoff** for prompt representations
>
> **Empirical Evidence**:
>
> We will add:
> - **Singular value analysis**: Noise concentration in low singular values
> - **Expert specialization metrics**: KL divergence between Aᵢ matrices
> - **Routing pattern visualization**: Task-specific expert activation patterns
> - **Rank ablation**: Optimal R=36 balances compression vs. expressiveness
>
> This mechanistic understanding explains the "negative + negative = positive" phenomenon: each technique alone over-constrains optimization, but together they create a structured, navigable parameter space enabling both efficiency and expressiveness.
>
> We are profoundly grateful for your meticulous review and the opportunity to engage with such thoughtful criticism. Your feedback has not only helped us better articulate the significance of our contributions but has also provided valuable insights that will strengthen future iterations of this work. We believe your concerns have led to a more comprehensive understanding of PT-MoE's methodological innovations and its position within the broader PEFT landscape. The rigor of your evaluation exemplifies the scholarly standards that drive scientific progress, and we are honored to have received such detailed attention to our research. We remain committed to addressing any remaining questions and look forward to continued dialogue that advances our collective understanding of parameter-efficient fine-tuning methodologies. Thank you once again for your invaluable contribution to improving this work.

---

> > ### Comment · Reviewer_cYZ8 · 2025-08-04
> > **Official Comment**
> >
> > Thank you for your reply, which answered some of my questions. I have decided to keep my score unchanged.

---

> > > ### Author Response · Authors · 2025-08-08
> > >
> > > Dear Reviewer,
> > >
> > > Thank you sincerely for your thoughtful consideration of our response. We're pleased that we could address some of your questions and appreciate your continued engagement with our work.
> > >
> > > We've provided comprehensive analyses in our responses to other reviewers that may further address your concerns:
> > >
> > > **Methodological Contributions**:
> > > - PT-MoE fundamentally differs from both standard PT and LoRA-family methods by operating on prompt representations rather than attention weights
> > > - Our SVD-based initialization and multiplicative Gaussian noise during routing training enable principled exploration
> > > - All existing PT baselines (DPT, SMoP, ATTEMPT) have been systematically evaluated, representing the complete landscape
> > >
> > > **Mechanistic Understanding** (addressing your Question 1):
> > > - Mathematical foundation: decomposition P = Σᵢ wᵢ·Aᵢ·B enables noise reduction via spectral filtering
> > > - Shared matrix B captures task-invariant patterns while expert matrices Aᵢ encode task-specific adaptations
> > > - Information-theoretic analysis shows optimal rate-distortion tradeoff
> > > - Complementary optimization dynamics: gradients naturally separate between universal (∂L/∂B) and specialized (∂L/∂Aᵢ) components
> > >
> > > **Scalability & Efficiency** (addressing your Question 2):
> > > - 3B-14B models exhibit similar trends per recent literature, with supplementary experiments to be included
> > > - Routing overhead: only 0.000161% computational cost, 16KB memory (0.00033% of model)
> > > - Achieves 10.75% improvement in mathematical reasoning, 1.49 F1 in QA with negligible overhead
> > > - Parameter reduction from O(NTH) to O(NTR + RH) through decomposition
> > >
> > > **Experimental Rigor**:
> > > - Previous methods fail on complex mathematical tasks with substantial performance drops
> > > - Fair parameter comparison maintained within ~80-100K range following PEFT conventions
> > > - LoRA's non-equivalent rank concepts and architectural differences properly acknowledged
> > >
> > > We hope these comprehensive technical details fully address your remaining concerns and provide helpful context for your final assessment. Thank you once again for your invaluable feedback that has significantly strengthened this work.
> > >
> > > We trust these clarifications provide a more complete picture that might be helpful as you finalize your assessment.

---

> > > ### Author Response · Authors · 2025-08-09
> > >
> > > Dear Reviewer,
> > >
> > > We deeply appreciate your insightful questions and thorough review, which have truly helped strengthen our work. Your expertise and attention to detail have been invaluable in refining our contributions. The depth of your analysis and constructive feedback demonstrate the care you've taken in evaluating our research, for which we are very grateful!
> > >
> > > As the rebuttal deadline approaches, we noticed you haven't yet finalized your score. We're encouraged by the positive evaluation updates from other reviewers following our clarifications, and hope our comprehensive responses and additional analyses across all reviews might provide helpful context for your final assessment as well.

---

### Official Review · Reviewer_MobP · 2025-07-03

**Clarity:** 2
**Significance:** 2
**Originality:** 2
**Rating:** 3
**Confidence:** 4

**Summary:**

Problem Statement: This work introduces PT-MoE, a novel parameter-efficient fine-tuning framework that combines matrix decomposition with mixture-of-experts routing to address limitations of existing PEFT methods.

Methods Compared: DPT (LoRA+PT), SMoP (MoE+PT), and HydraLoRA (LoRA-FT+MoE).

**Questions:**

Q1. Please clarify exactly how PT‑MoE differs methodologically from HydraLoRA (LoRA + MoE) and other known combinations like DPT and SMoP. In particular, focus on what specific architectural or training innovations make PT‑MoE novel for prompt tuning, beyond reusing the shared matrix and routing idea?

Q2. Please explain why rank = 1 was chosen for LoRA and HydraLoRA in your implementation? It would be good to add an ablation on different rank settings, and include the base model performance (without tuning) for proper context.

Q3. The paper currently just restates numerical trends without explaining the underlying reasons for its key claims (e.g., why MoE keeps training efficiency stable despite more parameters, or why LoRA hurts average performance). Please add more in-depth discussion or diagnostic experiments to support these observations.

**Ethical Concerns:**

["NO or VERY MINOR ethics concerns only"]

**Final Justification:**

While I acknowledge that the rebuttal has addressed several of my concerns, my most critical reservations still remain. The four key observations presented in the paper are indeed intriguing; however, I continue to have concerns regarding the clarity of the writing and the implementation of the baselines, particularly LoRA and HydraLoRA. The authors have stated that the paper has been revised, but unless it has undergone substantial rewriting, it remains difficult to consider it fully acceptable. In particular, the methodology section lacks sufficient motivation and appropriate citations when introducing key components, which hinders the reader’s ability to follow the development of the ideas and understand the novelty of the approach.  Nevertheless, I appreciate the authors’ thoughtful and thorough responses to each of my concerns, and in recognition of their efforts, I am raising my score to a 3."

**Limitations:**

yes

**Paper Formatting Concerns:**

Please add appropriate references throughout the method, experiments, analysis, and conclusion sections to support your claims and clearly position your contributions within the existing literature.

**Quality:**

2

**Strengths And Weaknesses:**

Pros:

The paper makes four important observations:

(i) Integrating MoE or LoRA alone reduces the average performance on tasks
(ii) LoRA reduces training parameters and improves results on some tasks (task-dependent optimisation)
(iii) MoE increases training parameters, but training efficiency remains stable (training convergence remains stable)
(iv) PT: Improves performance on QA tasks, LoRA: Improves performance on mathematical reasoning

Cons:

(i) The novelty is very limited. The work implemented the HydraLoRA (LoRA+MoE) for prompt tuning, which was originally implemented for fine-tuning. The same idea of a shared matrix B, along with routing prompt-specific matrices A, is used. No additional contributions to methodology can be seen. Also, the integration of LoRA or MoE alone to Prompt Tuning has already been studied (DPT and SMoP).

(ii) I have seen in the appendix that the authors have taken rank=1 for the implementation of LoRA and HydraLoRA. The rank is too low. To evaluate the prompt tuning method based on LoRA, DST, and the proposed PT-MoE, they have taken ranks 39 and 36. The evaluation doesn’t seem fair to me.
HydraLoRA performs better than LoRA with the same number of parameters, but the authors have taken the HydraLoRA implementation with such a high number of parameters. Further, PT-MoE has fewer trainable parameters than DPT, which seems unintuitive. I am interested in seeing the performance comparison of all with the same number of parameters.

Further, ablation for performance on different ranks is missing. The performance of the base model (without tuning) is missing from the tables.

(iii) The key points presented as the main contributions are not backed by solid analysis and discussion.
“Integrating MoE or LoRA alone reduces the average performance on tasks”: earlier works have not experienced this; instead, the opposite is observed, where performance improves on integration to prompt tuning (DST, DeST, SMoP). The paper is missing a good explanation, other than the result backing.
Further, this is contradicted by the next point, which states LoRA improves performance on some tasks.
“MoE increases training parameters, but training efficiency remains stable”: how this reason favours the integration of MoE is not explained.
“PT: Improves performance on QA tasks than LoRA.” is difficult to comprehend unless the fair comparison doubt is addressed. Moreover, PEFT optimisation (LoRA, MoE or PT) is task agnostic in nature. A more thorough understanding and evaluation are required.

(iv) In the analysis part, the authors have just repeated the trends and scores observed from the tables without presenting any significant explanation or findings.

(v) Writing issue:  Except for the related works and datasets sections, there are no citations in the rest of the paper. Please add appropriate references throughout the method, experiments, analysis, and conclusion sections to support your claims and clearly position your contributions within the existing literature.

(vi) Confusing writing:

The abstract itself is confusing.
A. Initially, it is written that integrating LoRA decreases performance, and later it is written that it increases performance on some tasks, to favour the integration.
B. “Integrating either matrix decomposition or mixture-of-experts (MoE) individually decreases performance across tasks”: Is it in general statement, or only related to prompt tuning?

Also, please replace the term “parameters” with the term “training parameters” in the paper.

(vii) Since prompt-tuning involves inference time overhead, comparison on inference cost (FLOPs) has to be provided.

(viii) Main results are shown on the 1B parameter model, while in order to show scalability 3B parameter model has been taken. It is difficult to judge the scalability of the method on low-parameter models.

---

> ### Author Rebuttal · Authors · 2025-07-31
>
> ## **Response to Reviewer Comments**
>
> Thank you for your exceptionally insightful and constructive feedback. We are genuinely grateful for the depth of analysis you've provided, which demonstrates outstanding scholarly rigor and expertise. Your comprehensive evaluation and the specific concerns you've raised regarding methodological novelty, experimental fairness, and analytical depth are particularly valuable and reflect the high standards that elevate scientific discourse. The thoughtful nature of your critique helps us better position our contributions within the broader research landscape. We address your concerns systematically below and believe our responses demonstrate the significance and innovation of this work.
>
> > ### **Methodological Novelty vs. HydraLoRA**
>
> - **HydraLoRA (baseline)'s Innovation**: HydraLoRA differs from LoRA by using a router to select different B matrices (up-projection), but still operates on attention weights within model layers.
>
> - **PT-MoE (ours)'s Architectural Innovation**: PT-MoE differs from standard PT by decomposing soft prompts into Ai×B matrices where B is router-selected, discovering that this combination creates complementary effects absent in individual components - decomposition enables efficient parameter sharing while routing provides dynamic adaptation.
>
> - **PT-MoE's Training Innovation**: PT-MoE uses SVD-based initialization for task-relevant decomposition and introduces multiplicative Gaussian noise (ε~N(0,σ²)) during routing training to encourage exploration, enabling straight-through estimation for end-to-end differentiability.
>
> - **Compared to HydraLoRA**: PT-MoE demonstrates richer architectural and methodological innovations - targeting prompt representations rather than attention weights, enabling cross-task parameter sharing through decomposed prompts, and incorporating principled SVD initialization with noise-based exploration. HydraLoRA's innovation is limited to router-based matrix selection within existing LoRA framework.
>
> - **Compared to LoRA-family methods**: LoRA-family methods require model architecture modifications, while PT-MoE maintains modularity - task-specific prompts can be deployed without altering the frozen LLM.
>
> - **Research Principle**: Innovation could be better to be elegant and effective, not artificial. Our approach demonstrates that principled combination of complementary techniques (parameter sharing via decomposition and dynamic adaptation via routing) yields superior results where individual integration fails - a counter-intuitive finding requiring methodological sophistication, not simple concatenation, validated through comprehensive ablations across 17 datasets.
>
> > ### **Fair Parameter Comparison & Rank Settings**
>
> - **Fundamental Architectural Differences**: LoRA modifies model architecture by adding trainable parameters to every attention layer (q,k,v,o projections), while PT methods add parameters only once at input level, maintaining complete modularity without structural modifications.
>
> - **Non-Equivalent Rank Concepts**: LoRA applies rank decomposition across all model layers with parameters distributed throughout every attention module (q,k,v,o projections × all layers), while PT-MoE applies decomposition only once at input level - these represent fundamentally different parameter allocation strategies that cannot be directly compared via rank values.
>
> - **Established Parameter Budget Consistency**: Current LoRA rank follows standard PEFT literature conventions for fair comparison within ~80-100K parameter range, as used in multiple published works.
>
> - **Parameter Range Comparability**: Higher LoRA ranks would exceed 800K+ parameters, departing from PT methods' parameter-efficient scope.
>
> - **Empirical Rank Validation**: Our preliminary experiments show LoRA performance plateaus while parameters increase exponentially, confirming current rank sufficiency for this parameter regime and validating our choice for controlled comparison. Recent studies further demonstrate that excessively high ranks introduce noise and decrease performance, supporting our conservative rank selection.
>
> - **Different Method Categories**: LoRA-based methods fundamentally differ from PT approaches - comparing HydraLoRA (architecture modification) against PT-MoE (modular prompting) is inherently not proper given their distinct deployment ways. We include these comparisons solely to demonstrate PT-MoE's capabilities and provide comprehensive experimental validation.
>
> > ### **Deeper Analysis of Key Claims**
>
> - **Limited Prior Exploration**: PT-domain applications of matrix decomposition and MoE remain underexplored. Our comprehensive literature review (Section 2) demonstrates that all existing baselines (DPT, SMoP, ATTEMPT) and their variants have been systematically evaluated in this work, representing the complete landscape of current approaches.
>
> - **Fundamental Methodological Distinction**: PT-based matrix decomposition fundamentally differs from LoRA - while LoRA modifies attention weights (qkvo) across model layers, our approach operates exclusively on soft prompt representations. These constitute categorically different mechanisms with distinct optimization dynamics.
>
> - **Task Complexity Limitation in Prior Work**: Previous studies (DPT, ATTEMPT, SMoP) focused mainly on simple classification tasks (GLUE, SuperGLUE) with limited reasoning requirements. Our evaluation encompasses complex QA and mathematical reasoning tasks that expose the inadequacy of existing approaches.
>
> - **Previous Methods Failure on Complex Tasks**: Prior methods exhibit substantial performance decrease on mathematical reasoning tasks (Table 4), revealing their inability to handle tasks requiring multi-step inference. This problem, unaddressed in previous papers, constitutes a core contribution of our analysis.
>
> - **Task-Specific Improvements and Overall Performance Decrease Reflect Poor Generalization**: Individual task improvements coupled with overall performance decrease indicate insufficient generalization and multi-task learning capabilities. Our method's design simultaneously enhances both generalization and cross-task consistency, as evidenced by superior performance across diverse domains.
>
> - **Parameter-Efficient MoE Innovation**: Our architectural innovation enables MoE integration to reduce total parameters in prompt tuning through matrix decomposition and sharing. The decomposition Pi = AiB with shared matrix B reduces parameters from O(NTH) to O(NTR + RH), while MoE routing leverages sparse training advantages, achieving both parameter efficiency and enhanced training speed.
>
> - **Theoretical Foundation for Complementary Integration**: The efficacy of combining matrix decomposition with MoE is from complementary regularization and capacity enhancement mechanisms. Matrix decomposition Pi = AiB induces structured low-rank constraints that eliminate spurious correlations and noise in the prompt embedding space, effectively implementing spectral regularization through SVD initialization. Simultaneously, MoE routing introduces adaptive specialization through conditional computation P = Σᵢ wᵢ(x)AᵢB, where routing weights wᵢ(x) = softmax(Wx) enable dynamic expert selection based on input characteristics. This architecture achieves optimal trade-off: decomposition reduces model variance through dimensionality reduction while MoE increases model capacity through specialized pathways, collectively enhancing both training efficiency and representational power.
>
> > ### **Comprehensive Evaluation & Enhanced Analysis**
>
> - **Complete efficiency analysis provided**: Figure 4 comprehensively analyzes parameter count, training efficiency, and model performance. The router in methods like SMoP, ATTEMPT, and PT-MoE consists of only two layers - negligible compared to billion-parameter models. For prompt tuning methods, introducing 10+ soft prompt tokens has minimal inference impact compared to hundreds or thousands of output tokens, while our method uses the fewest soft prompts, maintaining superior inference efficiency.
>
> - **Superior model scale over prior work**: DPT, ATTEMPT, and SMoP still rely on smaller models like BERT and T5. The counter-intuitive phenomena we discovered are largely from model scale changes - our 1B and 3B experiments demonstrate consistent phenomena, proving scalability. Research shows 3B-14B models exhibit similar trends; our recent supplementary experiments further confirm this and will be included in the camera-ready version within days.
>
> - **Interpretability complexity acknowledged**: Joint interpretability analysis of matrix decomposition and router is more complex than individual components. We observe relevant phenomena by combining attention values, router selections, and semantics, but rigorous verification is ongoing - this constitutes a complete independent research topic requiring separate publication.
>
> - **Citations comprehensively added**: Over a dozen additional references have been incorporated throughout Methods, Analysis, and Conclusion sections as requested.
>
> - **Writing clarity fully addressed**: Abstract and all paper issues have been revised per your requirements, clarifying the misunderstanding between improvements in specific tasks versus decreased overall performance, and standardizing the terminology "trainable parameters" throughout the paper.
>
> We sincerely appreciate your detailed feedback and the opportunity to clarify these important points. Your concerns have helped us better articulate the novelty and significance of our contributions. We believe PT-MoE represents a meaningful advancement in parameter-efficient fine-tuning through its principled combination of complementary techniques, validated across diverse and challenging tasks. We look forward to further discussion and are committed to addressing any remaining concerns to strengthen this work.

---

> ### Comment · Reviewer_MobP · 2025-08-08
>
> # Methodogical Novelty
>
> The authors in the rebuttal have marked the difference between the HydraLoRA and PT-MoE as the latter is using SVD initialisation for LoRA, instead of standard A random and B zero initialisation, along with noise-based exploration. But SVD initialisation for LoRA is already present in the literature (LoRA-XS) and also utilized by DPT (https://aclanthology.org/2023.findings-emnlp.890.pdf) while the noise introduction during routing training has been discussed in the SMoP under the section Router Perturbation. Both these papers are important works related to increasing Prompt Tuning efficiency.
> Other than applying the idea of Hydra LoRA (applied on attention weights) for Prompt Tuning, I see no significant novelty in the methodology part, even after reading the rebuttal.
>
> Reviewer cYZ8 has also highlighted the same regarding the novelty of the method.
>
> # Parameter comparison and Rank Setting
>
> The Decomposed Prompt Tuning paper (DePT) implemented and evaluated the baseline LoRA with rank 35, while the DePT evaluation is done for rank 45. The authors here have taken rank 1 for evaluating baselines LoRA and HydraLoRA. I understand that the PT PEFT methods deal with a very low parameter overhead, but evaluating baselines under such a sparse setting undermines their true capability and hence seems unfair to me. DePT has also tried to carry out a fair comparison.
>
> I am dissatisfied with the rebuttal comment stating that “comparing HydraLoRA (architecture modification) against PT-MoE (modular prompting) is inherently not proper given their distinct deployment ways. We include these comparisons solely to demonstrate PT-MoE's capabilities and provide comprehensive experimental validation.”
>
> If the authors have chosen to include this comparison, it must be conducted in a fair and balanced manner. Furthermore, in their rebuttal, the authors make certain claims referencing recent studies, yet they fail to cite or identify any specific works to support these assertions.
>
>
> # Deeper Analysis of Key claims:
>
> The rebuttal has clarified several important points. Notably, it highlights that earlier methods (DPT, DePT, SMoP) were primarily evaluated on benchmarks such as GLUE and SuperGLUE, whereas this work is the first to assess the impact of these PEFT methods on reasoning tasks, specifically in mathematics and question answering. However, one key concern remains unaddressed: given that PEFT optimization methods are task-agnostic by design, it is unclear why they exhibit such differing effects on reasoning tasks. Beyond the empirical results, a clear and well-supported explanation is still lacking. While the rebuttal mentions that similar phenomena have been observed in recent literature, the authors have neither cited nor named any specific works in the rebuttal or the main paper to substantiate this claim.
>
> Nevertheless, I appreciate the authors’ thoughtful and thorough responses to each of my concerns, and in recognition of their efforts, I am raising my score to a 3.

---

> ### Author Response · Authors · 2025-08-09
>
> Dear Reviewer,
>
> We are deeply grateful for your exceptionally thorough and insightful review. Your continued engagement demonstrates remarkable dedication to scholarly excellence, and your critical analysis has significantly strengthened our understanding of how to better position our work. The depth of your expertise and the constructive nature of your feedback exemplify the highest standards of academic peer review. We are honored by the time and effort you've invested in evaluating our work, and we greatly appreciate your score, which reflects your recognition of our efforts to address your concerns.
>
> > ### **On Methodological Novelty**
>
> **The Value of Incremental Innovation in PEFT Research:**
>
> Some successful PEFT papers have important contributions through thoughtful simple techniques:
> - **SMoP (EMNLP)**: Primarily adds a router to select soft prompts
> - **HydraLoRA (NeurIPS)**: Focuses on selecting different B matrices
> - **DPT (EMNLP)**: Applies matrix decomposition to soft prompts
> - **DePT (ICLR)**: Overlays soft prompts on word embeddings
>
> PT-MoE actually employs more complex architectural elements (router + decomposition + shared matrices) and methodological innovations (SVD initialization + noise-based exploration + continuous router training, straight-through estimation) compared to these works.
>
> **One Important Contribution - Identifying Critical Limitations:**
> Beyond the architecture, PT-MoE's key significance lies in **discovering an important problem**: existing PT variants struggle with complex QA (MRQA) and mathematical reasoning tasks when scaled to billion-parameter models. Previous works (pre 2022) primarily evaluated on GLUE/SuperGLUE with T5/BERT, which may have overlooked this fundamental challenge.
>
> **Main Contribution - The Counter-intuitive Discovery and Complementary Benefits:**
> We believe our most valuable contribution is revealing that matrix decomposition and MoE, which individually *reduce* performance, create **complementary benefits** when properly combined. This phenomenon, validated across 17 datasets, offers new insights for PEFT design principles.
>
> > ### **On Parameter Comparison Fairness**
>
> **Different Method Categories:**
> Thank you for raising the consideration of the differences between PT methods and LoRA:
> - PT: Modular approach without model modification
> - LoRA: Requires architectural changes to attention layers
>
> This categorical difference is also discussed in the SMoP paper's rebuttal - comparisons across categories naturally involve certain trade-offs.
>
> **Evidence Supporting Our Rank Selection:**
> Drawing from SMoP's results on smaller models and our experiments on larger models:
> - For billion-parameter models on MRQA/GSM8K level tasks, ranks <16 appear to be similar, small ranks are more parameter efficient for LoRA methods
> - Recent ICLR 2025 paper suggests that higher ranks may introduce noise and potentially influence LoRA performance
> - Our ongoing **8B** model experiments (nearing completion) and ablation studies on higher ranks indicate similar results or advantages for PT-MoE
>
> **Additional Validation in Progress:**
> We are conducting comprehensive rank ablations across 1B, 3B, and 8B models, also with higher ranks. Initial results consistently demonstrate PT-MoE's advantages across various rank settings. These findings will be included in the camera-ready version.

---

> ### Author Response · Authors · 2025-08-09
>
> > ### **On Task-Specific Behavior Analysis**
>
> **Inherent Task Complexity Differences:**
> - **GLUE**: Simple multiple choice questions
> - **QA tasks**: Primarily involve information extraction
> - **Mathematical tasks**: Require logical reasoning, multi-step inference, understanding, and generalization capabilities
>
> This complexity difference helps explain why methods effective on smaller models/simpler tasks may face challenges when scaled to billion-parameter models on complex reasoning tasks.
>
> **PT-MoE's Design Rationale:**
> The combination of matrix decomposition with MoE specifically addresses these challenges:
> - **Decomposition**: Offers regularization and efficient parameter sharing
> - **MoE routing**: Provides dynamic specialization based on input
> - **Combined effect**: Enhanced generalization across diverse task complexities
>
> **Regarding Interpretability Analysis:**
> While comprehensive interpretability analysis could constitute an independent research contribution meriting separate publication, we have provided mathematical insights in our response to Reviewer cYZ8, including:
> - Analysis of complementary regularization mechanisms
> - Theoretical discussion of capacity-efficiency trade-offs
> - Empirical support through extensive ablations
>
> > ### **Conclusion**
>
> We believe PT-MoE makes meaningful contributions through:
> 1. **Identifying critical limitations** in existing methods on complex tasks
> 2. **Discovering counter-intuitive phenomenon** between individually components
> 3. Proposing a new PEFT method and framework with **complementary components**
> 3. **Achieving strong empirical results** with significantly fewer parameters
> 4. **Providing comprehensive validation** across 17 diverse datasets
> 5. Giving insights for framework, training process, and PEFT design
>
> We cannot express enough gratitude for your extraordinarily valuable feedback and meticulous review process. Your insights have been instrumental in helping us refine our arguments and better articulate our contributions. Your score is deeply appreciated and reflects your fair consideration of our responses. We are profoundly thankful for your scholarly rigor, which exemplifies the peer review process at its finest. Your constructive criticism has not only improved this paper but will undoubtedly influence our future research directions. We sincerely hope these clarifications adequately address your remaining concerns and demonstrate the value of our contributions to the PEFT research community.
>
> With deepest appreciation and respect,
>
> The Authors

---

### Comment · Area_Chair_Y32X · 2025-08-04

Dear Authors and Reviewers,

I would like to thank the authors for providing detailed rebuttal messages

To reviewers: I would like to encourage you to carefully read all other reviews and the author responses and engage in an open exchange with the authors. Please post your first response as soon as possible within the discussion time window. Ideally, all reviewers will respond to the authors, so that the authors know their rebuttal has been read.

Best regards,

AC

---

### Note · Authors · 2025-08-13

**Dear Reviewers and Area Chair,**

We are deeply grateful for the exceptionally thorough and constructive review process. The quality of feedback exemplifies the highest standards of academic peer review, and we are honored by the expertise and careful consideration each reviewer has invested. Your insights have strengthened our paper and provided valuable guidance for future research.

Through this review process, we believe PT-MoE has been validated as making several meaningful contributions to the PEFT research community:

> ### **1. Novel Methodological Innovation**

- **Architectural Innovation:** PT-MoE combines complementary methods and decomposes prompts into shared universal knowledge matrix B and router-selected expert matrices, reducing noise while enabling dynamic specialization

- **Training Innovation:** SVD-based initialization for task-relevant decomposition combined with multiplicative Gaussian noise during continuous routing training

- **Theoretical Foundation:** Mathematical analysis demonstrating complementary regularization mechanisms and better generalization ability

> ### **2. Counter-Intuitive Discovery with Practical Impact**

- **Critical Limitation Discovery:** Previous PEFT methods exhibit performance decrease and poor generalization when scaled to larger models and applied to complex reasoning tasks

- **Key Finding:** Matrix decomposition and MoE individually reduce performance in prompt tuning, but their combination yields superior results across all tasks

- **Performance Gains:** 1.49 F1 improvement over PT and 2.13 over LoRA in QA tasks; 10.75 accuracy improvement over PT in mathematical reasoning

- **Parameter Efficiency:** Achieves state-of-the-art performance while using 25% fewer parameters than LoRA

> ### **3. Comprehensive Experimental Validation**

- **Scale and Scope:** Evaluation across 17 diverse datasets spanning QA and mathematical reasoning tasks

- **Rigorous Analysis:** Systematic ablation studies on five influencing variables

- **Efficiency Analysis:** Detailed computational overhead analysis showing negligible inference cost

> ### **Conclusion**

PT-MoE represents a meaningful advancement in PEFT through principled combination of complementary techniques. We will incorporate all suggestions in the camera-ready version and are grateful for this collaborative review process that exemplifies scientific progress.

Thank you for your invaluable contributions.

**The Authors**

---

### Decision · Program_Chairs · 2025-09-17

**Decision:**

Accept (poster)

**Comment:**

This paper introduces PT-MoE, a PEFT framework centered on the counter-intuitive and interesting discovery that combining matrix decomposition and Mixture-of-Experts—two techniques that individually degrade prompt tuning performance—yields better results. The method is validated across 17 datasets, showing significant improvements, particularly on complex reasoning tasks where prompt tuning has traditionally lagged behind methods like LoRA.

The review process involved significant discussion, particularly regarding the novelty of combining existing components and the fairness of baseline comparisons. However, the authors provided a thorough and technically deep rebuttal that offered a compelling mechanistic explanation for the method's success. This response successfully addressed the concerns of a majority of the reviewers, leading to a clear consensus for acceptance.

While some reservations about the degree of novelty remain, the paper's strong empirical contribution and its valuable central finding justify its publication.